# Preparation and Characterization of Waterborne UV Lacquer Product Modified by Zinc Oxide with Flower Shape

**DOI:** 10.3390/polym12030668

**Published:** 2020-03-17

**Authors:** Yan Wu, Xinyu Wu, Feng Yang, Jiaoyou Ye

**Affiliations:** 1College of Furnishings and Industrial Design, Nanjing Forestry University, Nanjing 210037, China; aprilwu2019@163.com; 2Co-Innovation Center of Efficient Processing and Utilization of Forest Resources, Nanjing Forestry University, Nanjing 210037, China; 3Fashion Accessory Art and Engineering College, Beijing Institute of Fashion Technology, Beijing 100029, China; 4DeHua TB New Decoration Material Co. Ltd., Deqing 313200, China; yejiaoyou2020@163.com

**Keywords:** poplar wood, waterborne UV lacquer product, wood modification, contact angle, spectroscopy, super-hydrophobic coating

## Abstract

In this paper, the waterborne UV lacquer product (WUV) was used as the main raw material, zinc oxide (ZnO) was used as the additive, and the stearic acid as the surface modifier. According to the method of spraying coating on the surface of poplar wood (*Populus tomentosa)*, a simple and efficient preparation method was carried out to generate a super-hydrophobic surface and enhance the erosion resistance of the coating. By testing, the contact angle (CA) of water on the coating surface can reach 158.4°. The microstructure and chemical composition of the surface of coatings were studied by scanning electron microscope (SEM), Fourier transform infrared spectroscopy (FT-IR), and X-ray diffraction (XRD). The results showed that under acidic conditions, the non-polar long chain alkyl group of stearic acid vapor molecule reacted with the hydroxyl group in acetic acid, the metal ions of the ZnO were displaced to the stearic acid and generated globular zinc stearate (C_36_H_70_O_4_Zn). The hydrophobic groups –CH_3_ were grafted to the surface of zinc stearate (ZnSt2) particles and the micro/nano level of multistage flower zinc stearate coarse structure was successfully constructed on the surface of poplar wood, which endowed it with superhydrophobic properties. It is shown that the coating has good waterproof and erosion resistance.

## 1. Introduction

Waterborne UV lacquer product (WUV) is a kind of coating which realizes crosslinking curable by UV irradiation [1]. It does not contain volatile toxic substances or irritating gases [2,3,4]. WUV combines UV-cured technology with waterborne polymer technology [5,6], which not only saves energy and protects environmental but also has the advantages of fast curing speed, pipeline production and high production efficiency [7,8]. The main component of WUV is waterborne acrylic resin [9]. It has the advantages of light color, high solid content, and strong adhesion, but its hardness, wear resistance, and mechanical properties limit its use [10,11,12]. However, due to the water-based acrylic resin with water as the dispersion medium, the phenomenon of incomplete curing is easy to occur during curing, and the residual water-based additives lead to poor water resistance of the coating. In addition, the evaporation of water requires more heat, which asks higher requirements during the drying process of the coating. In order to better play the application of UV cured coatings in the field of wood, the ultra-hydrophobic modification [13] and durability enhancement [14] are needed to improve the properties of WUV, so as to better meet the production and use requirements. Zhong et al. [15] used maleic anhydride and silicon modified waterborne alkyd resin, and prepared fluoro-acrylate resin by micro-emulsion polymerization without surfactant. The modified resin has better mechanical stability and corrosion resistance due to its larger contact angle with water. Saladino et al. prepared for the silica/PMMA nanocomposites with different silica quantities by a melt compounding method and systematically investigated it as a function of silica amount from 1 to 5 wt %. Results showed that silica nanoparticles are well dispersed in the polymeric matrix whose structure remains amorphous. The degradation of the polymer occurs at higher temperature in the presence of silica because of the interaction between the two components [16]. 

As a kind of inorganic material, zinc oxide nanoparticles have diversified morphology and excellent physicochemical properties [17]. Therefore, ZnO nanoparticles are a common packing, they are low cost [18,19,20], harmless to the environment [19], and have excellent photoelectric performance and rich form. Therefore, they have a broad application prospect in coatings [21,22,23], sensors [24,25], photoelectric material [26,27,28], medicine [20], and many other fields [29]. For example, Zahra et al. [30] used mixed ZnO/GO nanostructures to modify the surface of low carbon steel before acrylic resin coating. Through the structural properties and interactions between the oxygen-containing groups of ZnO and GO structures, the corrosion resistance of low carbon steel was improved. Zhou et al. [31] prepared the nano-hydroxyapatite/ZnO coating on biodegradable Mg-Zn-Ca block metallic glass by one-step hydrothermal method. Due to the presence of ZnO in the coating, the antibacterial rate of BMG in vitro was close to 100%. Guo et al. prepared two coatings on the surface of spruce panels, the first coating was coated by UV light absorbing ZnO and an additional hydrophobic layer of stearic acid, which endowed the wooden panels with water repellence as well as protected the ZnO coating from erosion due to rain. The second coating was based on a thin TiO_2_ layer attached to the wood surface, which aimed to avoid a pronounced initial color change induced by the coating itself [32]. Nair et al. prepared a stable dispersion of nanoparticles of three metal oxides, zinc oxide (ZnO), cerium oxide (CeO_2_), and titanium dioxide (TiO_2_) by propylene glycol (PG) through ultrasonication. The stability test of the coating was measured by UV–vis absorption spectroscopy and an accelerated weathering tester. Results shown that the increase in concentration of nanoparticles in the dispersion imparted higher resistance to UV induced degradation [33].

In wood science, the utilization of wood can be improved by changing its dimensional stability, flammability, biodegradability, and other properties [34]. Tuong et al. treated the acacia hybrid wood with TiO_2_ impregnation through the combination of pressure impregnation and hydrothermal post-treatment. Results showed that the color stability against UV irradiation of the TiO_2_ impregnated wood was significantly improved than that of the untreated acacia hybrid wood. The TiO_2_ nanoparticles were located on the inner surfaces of the wood vessels and it could improve the UV resistance of fabricated wood samples [35]. Yu et al. constructed ZnO nanostructures on the surface of solid wood via a simple two-step process consisting of generation of ZnO seeds on the wood surface followed by a solution treatment to promote crystal growth. Accelerated weathering was used to evaluate the photostability of treated wood. Results showed that the photostability of the treated wood was greatly enhanced [36].

The super-hydrophobic modification of WUV can make it have the properties of self-cleaning [37,38], antifouling [38] and water resistance [39] and improve the erosion resistance [3,40,41,42,43] of the coating, so that the wood material will possess a higher use value [44,45,46]. Yang et al. [47] soaked the poplar wood in a compound solution of maleic rosin and ethanol for 24 h. After being taken out and dried, the sample was then soaked again in the solution of TiO_2_ and modifier to obtain a super-hydrophobic wood with a contact angle up to 157°. After soaking the modified poplar in water for a week, irradiation under the hot sun for a week or boiling at 100 °C, the wood surface still has super hydrophobicity. Gao et al. prepared superhydrophilic and underwater superoleophobic poly by salt-induced phase inversion method. This kind of poly can quickly and efficiently separate the oil–water mixture system from the emulsified oil–water system, and the separation property is stable. After repeated use, the separation efficiency of the oil–water mixture was above 98%, and that of the oil–water emulsion is above 91%, which can be widely used in the oil–water mixture system and emulsified oil–water system [48]. Wang et al. [49] impregnated wood into a mixture of zinc acetate dihydrate and triethylamine, constructed a ZnO coating with roughness on the wood surface, and modified the coating surface with stearic acid to obtain a super-hydrophobic coating with low surface energy. The superhydrophobic coating remained superhydrophobic after drying at 60 °C for a month or soaking in deionized water for a week, showing good air stability and erosion resistance, which retained the natural appearance of wood while minimizing maintenance intervals. Poplar wood is a kind of fast growing wood, which is sustainable, biodegradable, biocompatible [34], rich in natural resources and low in price. It is usually used in furniture manufacturing. In this paper, ZnO nanoparticles (ZnO NPs) were used as the modifier to prepare for a superhydrophobic WUV coating on poplar wood, which retained the natural appearance of wood while minimizing maintenance intervals [32,50]. The possibility of large-scale application of this low-cost and environmentally friendly superhydrophobic surface preparation technology is discussed theoretically. This study is of guiding significance to the application of functional materials with superhydrophobicity.

## 2. Experimental Part

### 2.1. Materials

The poplar wood (*Populus tomentosa*) cut from the longitudinal section with the size of 20 mm long × 20 mm wide × 2 mm thick was supplied by Yihua Lifestyle Technology Co., Ltd., Shantou, China. Its moisture content was 9.9% and the absolute dry density was 298 kg/m^3.^ The above data was tested at 10 a.m. in winter, in an environment of 25 °C and the relative humidity of 46%. The waterborne UV lacquer product (A185721006) was purchased from Huzhou Dazhou Polymer Material Co., Ltd., Huzhou, China. Ethyl acetate (99.5%) and acetic acid (CH_3_COOH, 99.5%) were provided by Nanjing Chemical Reagent Co., Ltd., Nanjing, China. Zinc oxide (ZnO) was obtained from Xilong Scientific Co., Ltd.,Guangzhou, China. Anhydrous ethanol (99.7%), n-hexane (99%) and TBOT (98%) was purchased from Sinopharm Group, Shanghai, China. Stearic acid (C_18_H_36_O_2_) was provided from Yonghua Chemical Technology Co., Ltd, Changshu, China. 

### 2.2. Experimental Method

First, the poplar wood was dipped in anhydrous ethanol and distilled water respectively and experienced sonication by an ultrasonic crusher (Misonix, Inc., New York, USA) for 1 h, and then the treated wood was dried to absolute dry condition at 100 °C by a constant temperature blast drying oven (Shanghai Xinmiao Medical Devices Co., Ltd., Shanghai, China). 10 g WUV and 6 g ZnO powder was added into 13 mL ethyl acetate and the mixed solution was magnetically stirred by a magnetic stirrer (Yinyu High-tech Instrument factory, Gongyi, China) for 45 min, so that the WUV and ZnO powder were evenly dispersed in ethyl acetate. The obtained ZnO/WUV lacquer was sprayed onto dried poplar wood surface with a high-pressure electric spraying machine (Pritzker Power Tools, Ningbo, China) at a distance of about 15–20 cm. Two layers of UV lacquer product were applied to the poplar wood surface. The ZnO/WUV lacquer was cured under ultraviolet lamp (Wuxi Jinhua Test Equipment Co., Ltd, Wuxi, China) for 3.5 min and dried at 85 °C for 4 h to obtain ZnO/WUV coating.

Then the anhydrous ethanol was used as dispersant, 20 mL stearic acid and acetic acid were allocated as mixture under 70 °C. The solution was centrifuged at 4000 r/min with a centrifuge (Shanghai Anting Scientific Instrument Factory, Shanghai, China) for 5 min. The sonicated mixture was cooled to room temperature, and the ZnO/WUV sample was dipped into the mixture for 200 seconds before being removed. The multistage flower zinc stearate/waterborne UV lacquer super-hydrophobic coating (ZnSt2/WUV) was obtained after drying. The diagram of modification process of ZnO/WUV and ZnSt2/WUV coatings is shown in Figure 1.

### 2.3. Contact Angle (CA) Test

The contact angle of water on the cross section of ZnSt2/WUV was measured by Theta t200 Optical contact goniometer (Sweden baiorin technology Co. Ltd., Gothenburg, Swedish). After drying, the WUV and ZnSt2/WUV samples were placed on the loading platform. About 2 µL of deionized water was dripped on the surface of the samples under test. The sample was tested three times in parallel at different locations.

### 2.4. SEM

After the prepared WUV and ZnSt2/WUV samples were dried, thin slices with a width of about 5 mm and a thickness of about 2 mm were cut and fixed to the conductive adhesive on which the samples were placed with tweezers. The samples were sprayed with gold for 30 s with a vacuum plating apparatus. The surface and internal morphology of the samples were observed with an environmental QUANTA 200 SEM (FEI Company, Hillsboro, OR, USA) at 3 kV voltage.

### 2.5. FTIR and XRD

A Vertex 80V infrared spectrum analyzer (Germany Bruker Co., Ltd., Karlsruhe, Germany) was used to determine the functional groups of the ZnSt2/WUV, and the wave number range was set at 500–4000 cm^−1^. AXIS UltraDLD XRD (Nippon Koji Co. Ltd, Osaka, Japan) was used to characterize the internal molecular structure of the sample. The scanning speed was set from 5° to 70° [51], the acceleration voltage was 40 kV, and the impressed current was 30 mA.

### 2.6. Different pH Values of Solution and Organic Solvent Immersion Test

ZnSt2/WUV was soaked in the solutions with the pH values of 2, 4, 7, 9, 12, respectively and dried after a period of time [52]. Similarly, the samples were soaked in different organic solvents [53], dried after a period of time, and the contact angle of water on two samples in the same reagent were measured and the average value of them was obtained as the final data. 

### 2.7. Water Resistance Test

In order to determine whether the ZnSt2/WUV coating had good water resistance, two samples for each of WUV and ZnSt2/WUV were impregnated in distilled water, and then which were taken out and weighed after a period of time, and the water absorption before and after modification was calculated by using Equation (1):
(1)$$\mathrm{WA}\left(\%\right)=\left(m_{ai}-m_{bi}\right)/m_{bi}\times 100\%$$
where *m*_bi_ and *m*_ai_ represent the film weights before and after absorbed water, respectively [15].

## 3. Results and Discussion

### 3.1. Contact Angle (CA) Test

By testing, the contact angles of water on the WUV and ZnSt2/WUV (Table 1) coating samples were 68.2° and 158.4°, respectively. Compared with the WUV coating, the ZnSt2/WUV coating had the property of superhydrophobicity.

### 3.2. SEM

The surface microstructure of WUV and ZnSt2/WUV coating samples was shown in Figure 2. It could be seen from Figure 2A and 2B that the surface of WUV was smooth and covered by a uniform and continuous WUV, with a few nanoparticles distributed on the surface. It could be seen from Figure 2C that, at a magnification of 250, the surface of ZnSt2/WUV coating was evenly arranged with micron-sized bulbous flower clusters, indicating that under the adhesive effect of WUV and the modification effect of stearic acid, ZnO NPs reacted with acetic acid solution to form a micro/nano flower-like structure. As could be seen from the Figure 2D, the surface of the micron-scale bulge was evenly arranged with the petal-like mastoid structure. The petal-like mastoid tip had a nanoscale folded structure, which enabled the gas to exist in the structure and lift the droplets when contacting with the water droplets, so that the water droplets cannot penetrate and spread. The micro/nano structure, similar to that of rose petals, was the key reason to the superhydrophobicity of the ZnSt2/WUV coating [29,54].

### 3.3. FTIR and XRD

Figure 3A shows the FTIR spectrum of ZnSt2/WUV. 2869.5 cm^−1^ corresponded to the symmetric tensile vibration peak of C–H bond. 1320.9 and 1164.7 cm^−1^ were the asymmetric stretching vibration peaks of C–O–C, while 1488.7 and 1432.8 cm^−1^ were the bending vibration peaks of –CH_2_. 1376.9 cm^−1^ was the symmetric deformation vibration peak of –CH_3_. The absorption peaks corresponding to the reaction products –CH_2_ and –CH_3_ were 2923.5 and 2850.2 cm^−1^, respectively. It can be seen that the amount of C element and alkyl on the surface of ZnSt2/WUV increased significantly, indicating that the coating had been modified by stearic acid. The XRD pattern of the ZnSt2/WUV was shown in Figure 3B. It could be seen that the characteristic diffraction peak of zinc contained a small amount of C and O elements. In Figure 3B, the peaks at 16.2°, 22.6°, and 33.2° corresponded to the peaks of zinc stearate crystals. This indicated that the modification of stearic acid promotes the changes in topography of the surface and the formation of nano-particles, the carboxyl group in acetic acid reacted with ZnO particles, and the –CH_3_ hydrophobic group was grafted onto the surface of ZnSt2 particles, successfully constructing the superhydrophobic coating on the surface of poplar wood.

### 3.4. Different pH Values of Solution and Organic Solvent Immersion Test

The water contact angle of the water on ZnSt2/WUV coating was measured after impregnated in different pH values of solution and organic solvent. As could be seen from Figure 4A, the CA of coating was greater than 150° after being soaked in the solutions with pH values of 2, 4, and 7 for 50 h. After soaking in the solutions with pH values of 2, 4, and 7 for 100 h, the CA was greater than 140°, this was owing to the ZnSt2/WUV coating decompose into stearic acid and corresponding salt when it encounters acid, and its hydrophobicity would not lose, indicating its ability to maintain hydrophobicity in acid solution and neutral solution. The superhydrophobic property of the ZnSt2/WUV coating lost after impregnating in the solution with pH values of 9 for 15 h and 12 for 5 h, respectively. This indicated that it is difficult for ZnSt2/WUV coating to maintain hydrophobicity in alkaline solution for a long time. The contact angle was above 140° after the ZnSt2/WUV coating was impregnated for 60 h in the solution with pH value of 9 and the contact angle was above 140° after the coating was impregnated in the solution with pH value of 9 for 3 h. The contact angle of the coating was above 130° after immersion for 100 h in the solution with pH values of 9 and 12. The results showed that the surface of the sample was still hydrophobic after dipping with the alkaline solution, the hydrophobicity of ZnSt2/WUV can be well maintained in acidic medium. As shown in Figure 4B, because the ethyl alcohol and n-hexane used for impregnation was neutral and acetic acid was said to be acidic, the ZnSt2/WUV coating remained super-hydrophobic after being impregnated in the solution of n-ethane, anhydrous ethanol, and acetic acid for 50 h, respectively. The stability of ZnSt2/WUV coating in ethyl acetate and tetrabutyl titanate solutions—which all present alkaline after hydrolysis—was relatively poor, which might be due to the reason that the low surface energy material on the coating surface was easy to dissolve in alkaline organic solvent, resulting in the loss of superhydrophobicity of the coating. In Figure 4A, the R2 coefficient of the contact angle of water on ZnSt2/WUV with the immersion time is −0.93, and in Figure 4B, the R2 coefficient of the contact angle of water on ZnSt2/WUV with the immersion time is −0.94. It can be seen that there is a negative correlation between the contact angle of water on ZnSt2/WUV and the immersion time. In a word, the coating could be better to maintain its hydrophobicity in acid organic solvents. Therefore, the application of this superhydrophobic coating onto wood surface could protect wood from erosion when it is exposed to an acid or alkaline environment, and extend the use of poplar wood.

### 3.5. Water Resistance Test

The water resistance test results of WUV and ZnSt2/WUV coatings are shown in Figure 5. WUV and ZnSt2/WUV was soaked in distilled water for 45 days. The water absorption rate increased with the increase of days and the water absorption of WUV went up even more and faster. When the samples were immersed in water for 45 days, the absorption rate of the WUV coating increased from 6.1% to 88.9%, while that of the ZnSt2/WUV increased from 0.9% to 67.8%. It could be seen that the ZnSt2/WUV showed good water resistance. This is because the modification of stearic acid constructed a super-hydrophobic structure of multigrade ZnO on the surface of poplar wood, so that the wood surface containing hygroscopicity components was covered by ZnSt2, which reduced the hygroscopicity of poplar wood. According to the test, after 50 days of immersion in distilled water, the contact angle of water on the surface of the sample was 154.4°, which showed a persistent superhydrophobicity [53]. Therefore, this method of constructing superhydrophobic coatings on the surface of wood substrates was expected to expand the use of poplar wood in waterproof and stain resistant [55].

## 4. Conclusions

A superhydrophobic coating was constructed on the surface of poplar wood with a contact angle of up to 158.4° through the water-based UV-cured wood coating which was modified by ZnO and stearic acid. The results showed that under acid condition, the nonpolar long chain alkyl group of stearic acid vapor molecule reacted with the hydroxyl group in acetic acid, the metal ions of the ZnO were displaced to the stearic acid and generated globular zinc stearate (C_36_H_70_O_4_Zn). –CH_3_ hydrophobic groups were grafted to the surface of ZnSt2 particles and micro/nano level of multistage flower ZnSt2 coarse structure was successfully constructed on the surface of poplar wood, which was the key to the superhydrophobic property of the coating. The pH, corrosion resistance, and water resistance tests revealed that the ZnSt2/WUV coating had good resistance to acid medium and some organic solvent corrosion ability. Compared with WUV, the water resistance of ZnSt2/WUV was stronger, which was conducive to prepare superhydrophobic coatings in an easy and environmentally friendly way and expand the application scope of poplar products in waterproof field.

## Figures and Tables

**Figure 1 polymers-12-00668-f001:**
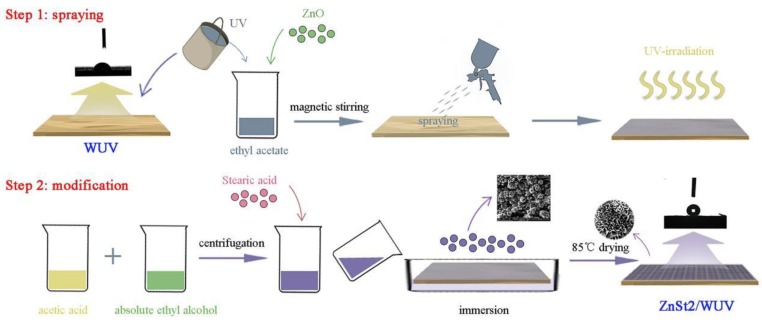
Diagram of modification process.

**Figure 2 polymers-12-00668-f002:**
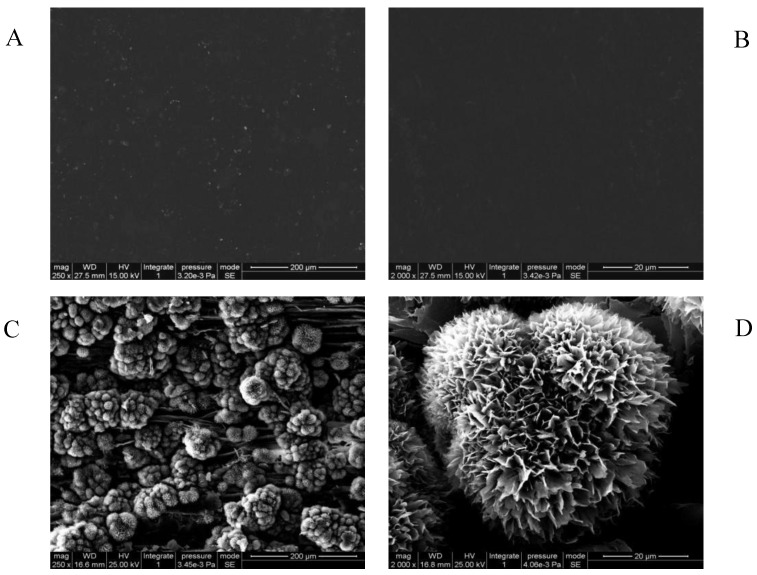
SEM images of WUV (**A**: at 250 magnification; **B**: at 2000 magnification) and ZnSt2/WUV (**C**: at 250 magnification; **D**: at 2500 magnification) coatings.

**Figure 3 polymers-12-00668-f003:**
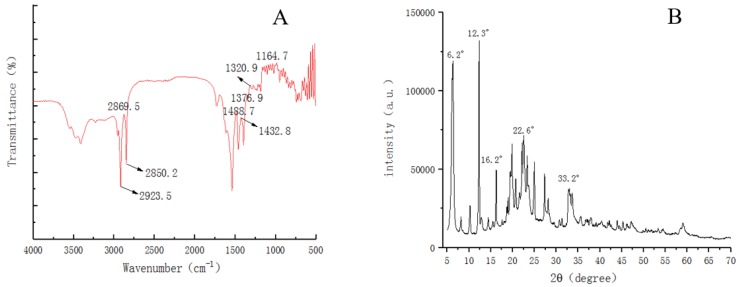
FTIR (**A**) and XRD (**B**) of ZnSt2/WUV.

**Figure 4 polymers-12-00668-f004:**
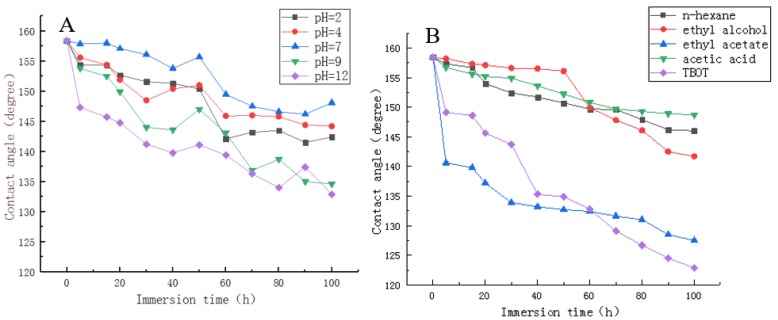
Different pH values of solution and organic solvent immersion test of ZnSt2/WUV (**A**: immersion in different pH solutions; **B**: immersion in different organic solvent).

**Figure 5 polymers-12-00668-f005:**
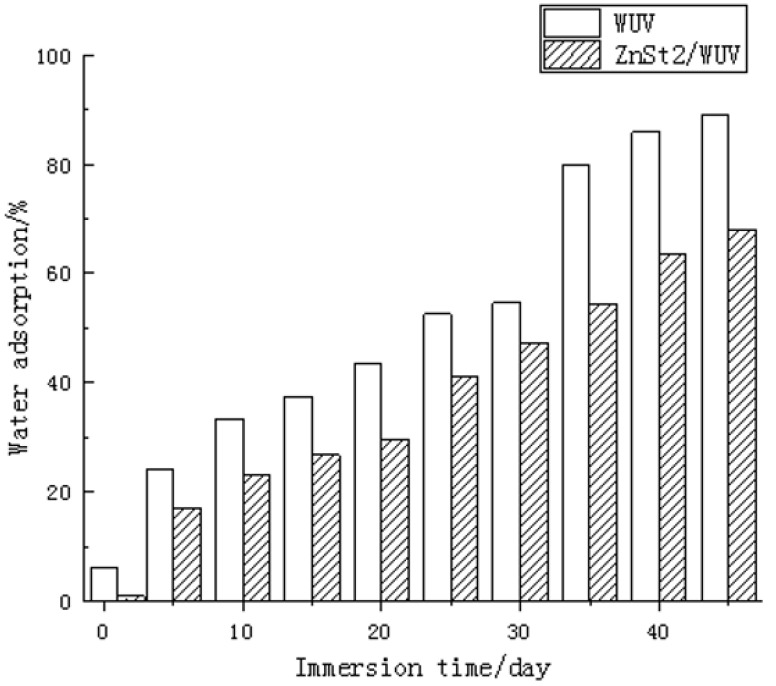
Water resistance test of WUV and ZnSt2/WUV.

**Table 1 polymers-12-00668-t001:** Image of contact angle (A: WUV; B: ZnSt2/WUV)

Name of the Coating	Contact Angle of Water on the Coating Surface (°)	Image of Contact Angle
WUV	68.2	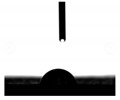
ZnSt2/WUV	158.4	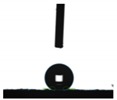

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
