# Peer review of "Preparation and Characterization of Waterborne UV Lacquer Product Modified by Zinc Oxide with Flower Shape"

_polymers, 2020, doi:10.3390/polym12030668_

Round 1

Reviewer 1 Report

Dear authors,

thank you for your revisions. A lot of work was done and now after changed meaning of some result reviewer can agree with achieved results.

With Best Regards.

Author Response

Dear Editors:

Thanks again to the reviewer's valuable comments on “Preparation and characterization of waterborne UV lacquer product modified by zinc oxide with flower shape”, it is our honor. We have sent the revised manuscript and have made changes by using the “track change mode”. The following are the point-by-point response and the corresponding revisions.

Reviewer: 1 
Dear authors,

thank you for your revisions. A lot of work was done and now after changed meaning of some result reviewer can agree with achieved results.

With Best Regards.

Answer: We sincerely thank you for your comments and suggestions for improving the quality of our article.

Thank you again for your valuable suggestions and it is our honor. We attach great importance to it and make corresponding modifications.

Thank you and best regards.

Yours sincerely,

Yan Wu and Xinyu Wu 

Reviewer 2 Report

LINE 32, 33 – delete “corrosion resistance”

LINE 64 – UV and not uv

LINE 150 – altrasonic or ultrasonic?

LINE 325 – add wood after poplar

Author Response

Dear Editors:

Thanks again to the reviewer's valuable comments on “Preparation and characterization of waterborne UV lacquer product modified by zinc oxide with flower shape”, it is our honor. We have sent the revised manuscript and have made changes by using the “track change mode”. The following are the point-by-point response and the corresponding revisions.

Reviewer: 2 

LINE 32, 33 – delete “corrosion resistance”

Answer: OK. It has been deleted.

LINE 64 – UV and not uv

Answer: Yes. It has been modified.

LINE 150 – altrasonic or ultrasonic?

Answer: Sorry, it is a mistake and has been clarified by “ultrasonic”.

LINE 325 – add wood after poplar

Answer: OK. All the statements in the manuscript have been corrected.

Thank you again for your valuable suggestions and it is our honor. We attach great importance to it and make corresponding modifications.

Thank you and best regards.

Yours sincerely,

Yan Wu and Xinyu Wu 

This manuscript is a resubmission of an earlier submission. The following is a list of the peer review reports and author responses from that submission.

Round 1

Reviewer 1 Report

GENERAL REMARK 1: The coating surface does not have a contact angle but a liquid on a coating has one. Therefore rephrase all parts where you are speaking of the contact angle of coating surface instead of contact angle of water on coating surface. It starts at the LINE 19, 117, 118 …

LINE 20 - delete space between the number and degree. In the same way correct the whole article (only for the degree of contact angle).  

LINE 23 – showed and not show

LINE 57 – nanoparticles are and not is

LINE 69 – “improve the durability” In which way or what kind of durability? Natural durability or durability against some other agent?

LINE 85 – again durability. When speaking of durability in connection to wood and coatings we always think on the durability (under natural exposure or artificial weathering) of a coating itself or on natural durability which is being prolonged due to the coating. In this paper nothing in this way was done, therefore please rephrase and avoid misuse of the term “durability”.

2.1 MATERIALS – It is not clear how broad the investigation was made. As it is written now, only 1 sample of defines size was used. Please add the info how sampling was made, how many samples for one treatment and how many for another, how many for a certain test one and how it was maintained that wood samples were homogenous/comparable between these sets.

Alo add the info on the growth ring orientation!

LINE 99 – “treated” Explain how.

LINE 104 – It is not clear how the samples were sprayed, only on one surface or all around?

LINE 106 – What do you mean by original?

LINE 109 – were instead of was

LINE 110 – “modify” Explain how.

LINE 112 – is instead of was

LINE 113 – Figure instead of figure

2.3 CONTACT ANGLE (CA) TEST – misuse of the contact angle and surface. Already explained in general remark 1. Also add in the text from which direction (cross section or somehow mention the grain direction) the contact angle was measured.

You are mentioning 2 litres of water for measurements. This is really a lot. Instead of used water volume name the volume of the droplet during the measurements of water on coating.

LINE 124 – were and not was

LINE 132 – delete “prepared above”

GENERAL REMARK 2: in the line 137 firstly appear the term “corrosion resistance test”. This is a misuse of this term. Corrosion testing or corrosion resistance testing (also known and referred to as “salt spray” testing) is an industry standard test used to check corrosion resistance of materials and surface coatings. Usually, the materials to be tested are metallic (although stone, ceramics, and polymers may also be tested) and finished with a surface coating which is intended to provide a degree of corrosion protection to the underlying metal. REPRHASE THROUGH THE HOLE PAPER AND AVOID THIS MISUSE.

LINE 139 – define a period of time!

2.7 WATER RESISTANCE TEST – Were samples immersed or impregnated (impregnation at a pressure of …)? How the samples were prepared? If the samples were sprayed by one side, were other sides of wood somehow sealed to prevent water uptake?

LINE 149 – use text editing instead of equation for Mbi and Mai and use m for a mass and not M as for weight.

3.1 CONTACT ANGLE – define what the deviation was during the measurements! The discussion in lines 155, 156, 157 should be moved to some place later since nothing of the mentioned stuff has at this stage in the results is shown jet. Also the discussion should be rephrased since here are only a speculations on anti-fouling and self-cleaning properties which were not directly monitored in this research.

LINE 168 – pear-like or peel-like? Please uniform.

FIGURE 3 – lighten the Figure 3A and 3B to increase the visibility.

LINE 177 shows instead of showed

3.3 FTIR AND XRD – The mentioned peaks are not shown on the Figure 4A, but some others are. Why? The XRD pattern is not properly explained. Please add additional how something is shown, by which peak and so on.

GENERAL REMARK 3: in the line 139 the pH stability of the coating is mentioned. This is again a misuse of this terms. A liquid has a pH and talking about how stable this pH of a liquid is is one thing and another thing is how a liquid of a certain pH affects or changes the contact angle of water on coating is another thing. REPHRASE WHOLE ARTICLE

LINE 212 – 100 % acetic acid is very acidic. What do you mean by “was said to be weakly acidic”. Explain and add the info on the % of solution for the acetic acid being used.

LINES 220, 221, 222 – the discussion should be rephrased. The connection to the service life prediction is too bold.

LINE 228 – are instead of were

LINE 231 – a good water resistance is mentioned. I do not agree with this statement since water resistance was lowered only for about ¼. And even all values are quite high, therefore a doubt in how the water adsorption was done arise.

LINE 234 – surface energy was not determined!

LINE 236 – the experiment was too narrow with not enough measurements and with very little samples that speculation on service life prediction is not possible. The same argument is valid for the discussion in the conclusion, LINE 252, 253.

Reviewer 2 Report

Dear Authors,

This topic of research is very interesting and important for improving of wood coatings durability.

You achieved very high hydrophobicity of treated surfaces.

But reviewer can not agree with some conclusions written in this paper.

Major comments:

In introduction more literature focused on similar topic (nanoparticles on wood) is missing: For example review paper Mishra et al. (2018) in Recent Patents on Nanotechnology, Sala et al (2012) in Polymer Degr. and Stab., Nair et al. (2018) in J. of Photochem. and Photobiol. B,  Guo et al. (2017) Informes de la Construction, Yu et al. (2010) Holzforschung, Tuong et al (2015) Bioresources, Gao et al. (2016) Chemical J. of Chinese University, and maybe others…

L44-47 – acrylic coatings with high solid content are used in industry – so this part is questionable…

Paper stated that this improvement can increase durability of coating in the interior and also exterior.

Part 3.4. is named corrosion resistance test.

But, by reviewer opinion and knowledge you have done only test against cold liquids (or chemicals) and water what say nothing about properties of coating in real conditions.

Based on Figure 3 it is possible to predict, that this surface will be very sensitive for mechanical abrasion in the interior during using and in the exterior during rain, wind and particle action.

And maybe after degradation of this structure properties of coating system could be also not better in comparison with simply acrylic coating system…

Based on this – other tests are required, to be possible to say sentences in L.29-30  or L 252-254. Minimum is abrasion test for indoor conditions and 12 weeks lasted artificial tests using UV and spraying by water (better is 2 years outdoor test) for prediction in outdoor conditions.  

Then could be interesting to measure again contact angle of surfaces – it was done in work Pánek et al. (2017) in Coatings and durability of hydrophobic surfaces was not very good after artificial weathering.

Minor comments:

In part 3.1. and 3.2. lotus leaf effect can be discussed using relevant citation from literature.

In Fig 5.  Reference WUV is missing – maybe it can have better results and lower proportional  decreasing of CA after action of chemicals?

Finally: Authors prepare superhydrophobic surface on coated wood (what is very nice result!) and do some tests against water and liquid. But it is not possible to say after achieved results if this coating system will be really more durable in real conditions.

In this form I suggest reject this paper, maybe after rewriting of some parts or doing of additional tests it can be accepted for publication.

Reviewer 3 Report

GENERAL COMMENT

The Authors analyzed a super-hydrophobic surface and enhance the wood durability by using UV lacquer product.  The authors undertook the realization of the current subject concerning with the proecological lacquer systems.

Paper requires some additions. Authors should answer the following questions and make changes in the text: 

Abstract

  • The Authors used the wrong term “UV cured coating”. The correct term is UV lacquer product. Only after hardening it can be received “coating”. 

Keywords

  • Authors should change and add keywords e.g. poplar wood (instead poplar), UV lacquer product (instead UV-cured coating), wood modification (instead super-hydrophobic modification), contact angle, spectroscopy, super-hydrophobic coating 

Introduction

  • Authors should explain why poplar wood was used for the tests.
  • The aim of the investigations has not been presented.
  • A disadvantage is the low number of papers of the foreign authors (e.g. from Europe). 

Experimental part

  • Authors in a simple way described investigation methods.
  • Authors didn’t give basic information about properties of the poplar wood (moisture content). The wood density in another unit (kg/m3) should be given. It is necessary from the wood science point of view.
  • What about conditioning parameters (time, temperature, relative humidity) of poplar wood prior to the measurement?
  • How many samples were used for each investigation?
  • The samples with the dimensions 20 mm long × 20 mm wide × 2 mm were used for tests. Were there any differences between the samples?
  • How many layers of the UV lacquer product were applied?

Results and Discussion

  • The Figure 2 is not necessary. It is better to give a table with the contact angle measurements with the basic statistic.
  • Authors wrote “…This indicated that the modification of stearic acid promotes the surface roughness and the formation of nano-particles…”. There is a wrong sentence. For such sentence more analysis is needed (e.g. roughness).
  • An important shortcoming is the lack of deeper spectra analysis. Probably it will be in next papers.
  • The curves on the Figure 5 with the mathematical formula and R2 coefficient should be described.
  • The units on the axes should be given in brackets.
  • There is a lack of the information about Figures in the text.

Conclusions

  • Authors should add the conclusion connected with the poplar wood as a substrate for the modification and finishing.
  • The name of the chapter should be “Conclusions” not “Conclusion”.

References

  • The authors quoted too many of their own works!

Paper can be published after minor changes.